# SGLT2 Inhibitors: Nephroprotective Efficacy and Side Effects

**DOI:** 10.3390/medicina55060268

**Published:** 2019-06-11

**Authors:** Carlo Garofalo, Silvio Borrelli, Maria Elena Liberti, Michele Andreucci, Giuseppe Conte, Roberto Minutolo, Michele Provenzano, Luca De Nicola

**Affiliations:** 1Division of Nephrology, University of Campania “Luigi Vanvitelli”, 80137 Naples, Italy; carlo.garofalo@unicampania.it (C.G.); dott.silvioborrelli@gmail.com (S.B.); m.elenaliberti@libero.it (M.E.L.); giuseppe.conte@unicampania.it (G.C.); roberto.minutolo@unicampania.it (R.M.); 2Division of Nephrology, University Magna Grecia, 88100 Catanzaro, Italy; andreucci@unicz.it (M.A.); michiprov@hotmail.it (M.P.)

**Keywords:** diabetes, chronic kidney disease, GFR, albuminuria, SGLT-2 inhibitors, end stage renal disease, survival

## Abstract

The burden of diabetic kidney disease (DKD) has increased worldwide in the last two decades. Besides the growth of diabetic population, the main contributors to this phenomenon are the absence of novel nephroprotective drugs and the limited efficacy of those currently available, that is, the inhibitors of renin-angiotensin system. Nephroprotection in DKD therefore remains a major unmet need. Three recent trials testing effectiveness of sodium-glucose cotransporter 2 inhibitors (SGLT2-i) have produced great expectations on this therapy by consistently evidencing positive effects on hyperglycemia control, and more importantly, on the cardiovascular outcome of type 2 diabetes mellitus. Notably, these trials also disclosed nephroprotective effects when renal outcomes (glomerular filtration rate and albuminuria) were analyzed as secondary endpoints. On the other hand, the use of SGLT2-i can be potentially associated with some adverse effects. However, the balance between positive and negative effects is in favor of the former. The recent results of Canagliflozin and Renal Endpoints in Diabetes with Established Nephropathy Clinical Evaluation Study and of other trials specifically testing these drugs in the population with chronic kidney disease, either diabetic or non-diabetic, do contribute to further improving our knowledge of these antihyperglycemic drugs. Here, we review the current state of the art of SGLT2-i by addressing all aspects of therapy, from the pathophysiological basis to clinical effectiveness.

## 1. Introduction

In the last quarter century, the incidence and prevalence of chronic kidney disease (CKD) have increased significantly, by 89% and 87%, respectively; more importantly, death due to CKD has risen sharply by 98%, with an increment of disability-adjusted life years of 92% [1]. These changes are mainly related to the increase of diabetes mellitus (DM), which has become the major cause of the global burden of CKD [2].

Notably, incidence rates of major cardiovascular (CV) diabetic complications, that is, acute myocardial infarction, stroke, and lower-extremity amputation, markedly declined between 1990 and 2010, while only a small decline was observed in the occurrence of end-stage kidney disease (ESKD) [3]. Similarly, in NHANES surveys from 1998 to 2014, the prevalence of CKD in DM patients remained stable at around 30% [4]. The discrepancy between the rates of CV and renal complications is likely related to the absence of new nephroprotective drugs. Indeed, no novel agent has been introduced in clinical practice since renin-angiotensin system (RAS) inhibitors almost 20 years ago, at the beginning of the 2000s [5,6]. Notably, the nephroprotective efficacy of RAS blockers remains unsatisfactory in patients with Diabetic Kidney Disease (DKD), leaving more than 40% of treated patients at high risk of progressing toward ESKD [5,6]. Similarly, achieving adequate glycemic control only attenuates the risk of new onset DKD [7,8].

Novel therapeutic strategies aimed at improving renal prognosis in diabetic patients are therefore urgently needed. This claim becomes even more crucial when considering the high incidence (40%) of new-onset CKD among individuals with type 2 DM (DM2) [9,10], as well as the high mortality risk in subjects with established DKD [10,11,12,13]. In this regard, sodium-glucose cotransporter 2 inhibitors (SGLT2-i) appear to be a viable therapeutic tool as they combine improvement of glycemia control with nephroprotective effects [14].

## 2. Pathophysiology of Glucose Handling in the Kidney and ROLE of SGLT2 Inhibition

Under conditions of normoglycemia or mild hyperglycemia, the renal proximal tubules reabsorb almost the whole amount of filtered glucose by sodium-glucose cotransporters [15,16]. SGLT2 are located in the early proximal tubule and are responsible for 90% of reabsorption of filtered glucose [17]; type 1 of these cotransporters (SGLT1) reabsorbs the remaining 10% in the late proximal tubule [16]. Glycosuria appears when glycemia exceeds the threshold level of 180 mg/dL with inter-individual variability [18,19]. In diabetic status, hyperglycemia is perpetuated due to the increase of the threshold for glycosuria to 200–240 mg/dL. In the presence of diabetes, hyperglycemia (directly) and enhanced intrarenal synthesis of Angiotensin II (indirectly) promote the growth of proximal tubules with a concomitant increased expression of SGLTs [14,20,21]. The enhanced expression of SGLT proteins and mRNA, also demonstrated in human tubular cells [22,23,24], contributes to the increased glucose reabsorption observed in diabetic patients (Figure 1).

Several biological mechanisms could explain the nephroprotective effects of SGLT2 inhibitors (SGLT2-i). Improvement of glycemic control is only one determinant of nephroprotection correlated with the use of SGLT2-i. A major mechanism independent of glycemia is the restoration of normal tubulo-glomerular feedback (TGF) (Figure 1). Indeed, glomerular hyperfiltration is a hemodynamic change that affects about 75% and 40% of type 1 and type 2 DM, respectively, at the beginning of disease [25,26,27,28].

The “physiological goal” of TGF is to limit salt dispersion (and dependent volume depletion) in the presence of high blood pressure, perfusion in the renal artery, or reduced proximal tubular reabsorption. This mechanism is constantly active, with the extent of the effect (increased tone of afferent arteriole due to adenosine release) being dependent on the delivery of sodium chloride to the macula densa. In diabetes, SGLT2-dependent higher sodium reabsorption (which reduces sodium delivery to macula densa) leads to afferent vasodilation and dependent glomerular hyperfiltration. SGLT2-i, by limiting the abnormal reabsorption of sodium chloride in the proximal tubule, restores a normal TGF activity and glomerular filtration (Figure 1).

In addition to the effects mediated by intrarenal hemodynamic changes, SGLT2-i also have direct anti-inflammatory and antifibrotic nephroprotective effects. Indeed, SGLT2-i suppress the production of reactive oxygen species, lessening glomerulosclerosis and tubulo-interstitial fibrosis [29,30]. Finally, an emerging additional effect of these drugs is the improvement in muscle-specific insulin sensitivity [31]. This effect has been confirmed in treated diabetic patients [32,33,34]. This observation is physiologically consistent with the loss of calories and resulting decrease of body weight. Notably, CKD *per se*, that is, even in the absence of diabetes, leads to insulin resistance; in fact, CKD is state of heightened inflammation with elevated levels of pro-inflammatory cytokines, and it is also associated with metabolic acidosis, excessive aldosterone and angiotensin II levels, accumulation of urea and uremic toxin, that is, all factors that promote insulin resistance [35].

## 3. SGLT2 Inhibitors for Cardio-Renal Protection in DM2: Randomized Controlled Trials

Three large trials have shown improvement in CV outcome, and renal prognosis as secondary outcome, in over 34,000 participants with DM2 at high CV risk [35,36,37,38,39,40]: EMPA-REG (Empagliflozin Cardiovascular Outcome Event Trial in Type 2 Diabetes Mellitus Patients); CANVAS (Canagliflozin Cardiovascular Assessment Study); DECLARE-TIMI-58 (Dapagliflozin Effect on Cardiovascular Events–Thrombolysis in Myocardial Infarction 58) (Table 1 and Table 2).

EMPA-REG, CANVAS and DECLARE trials consistently demonstrated a significant reduction of CV risk in the SGLT2-i arm, in particular in terms of lower rates of hospitalization for heart failure (HF). Renal events, tested as secondary endpoints, were also generally lowered by these drugs. Reduction in albuminuria and preservation of estimated Glomerular Filtration Rate (eGFR) appear to be a class-effect of SGLT2-i. In EMPA-REG, empaglifozin reduced the progression to severely increased albuminuria by 38% compared with placebo [37]. Similarly, CANVAS showed a 27% reduction of relative risk (RR) with canagliflozin for albuminuria progression [38]. These beneficial effects hold true also when examining smaller studies. In a double blind, randomized, crossover trial enrolling 33 diabetic patients with baseline albumin/creatinine ratio >100 mg/g, dapagliflozin decreased 24-h urine albumin excretion by 36% (*P* < 0.001) compared to placebo [41]. Furthermore, patients with stage 3 CKD also showed regression to a lower albuminuria category when treated with dapaglifozin [42]. In terms of eGFR, SGLT2-i produce a small reduction of eGFR at the beginning of treatment; these acute and transitory decline translates to a more preserved eGFR over the long-term independently from glycemic control [37,41,42,43,44].

Although the findings from EMPA-REG, CANVAS and DECLARE have provided signals on the nephroprotective properties of SGLT2-i in DKD, clinical trials primarily designed to evaluate outcomes in large CKD population are still awaited. Interestingly, two trials will evaluate the effect of SGLT2-i on renal outcome of patients with non-diabetic CKD: A Study to Evaluate the Effect of Dapagliflozin on Renal Outcomes and Cardiovascular Mortality in Patients With CKD (Dapa-CKD-Clinical-Trials.org identifier NCT03036150), and The Study of Heart and Kidney Protection With Empagliflozin (EMPA-KIDNEY, Clinical-Trials.org identifier NCT03594110).

Data specific to the diabetic population with CKD (low eGFR and pathologic albuminuria) have been provided for the first time by the CREDENCE trial (Canagliflozin and Renal Endpoints in Diabetes with Established Nephropathy Clinical Evaluation) [45]. CREDENCE has been prematurely stopped after 2.6 years because interim analysis indicated that the primary outcome had been met. Results demonstrate that in the DKD population at high risk of progression to ESKD, canaglifozin added on the top of optimal nephroprotective therapy (including anti-RAS at maximal tolerated dose in all patients) allows a 30% lower risk of the primary endpoint (composite of ESKD, that is, dialysis for at least 30 days, transplantation, or a sustained eGFR of less than 15 mL/min/1.73 m^2^ for 30 days, doubling of the serum creatinine for at least 30 days, or death from renal or cardiovascular disease). Some key comments must be made on the nephroprotective efficacy of canaglifozin shown in CREDENCE. First, the nephroprotective effect was maintained also in the subgroup with more advanced CKD (Hazard Ratio, HR, for primary endpoint was 0.75; 95% confidence interval, CI, 0.59–0.95 in the eGFR stratum 30–45 mL/min/1.73 m^2^); this observation is especially important for nephrologists that treat more severe renal disease. The second aspect to be highlighted is the effect of canaglifozin on a solid marker of progression to dialysis, namely the doubling of serum creatinine; for this endpoint, the HR was even more reduced (0.60, 95% CI, 0.48–0.76, *P* < 0.001) with respect to the primary combined endpoint. Third, the nephroprotective effect was anticipated by the sustained 31% reduction in albuminuria in the canaglifozin arm, in evidence from the 6-month control visit after randomization, thus confirming the well-known predictive value of albuminuria change on long-term renal outcome. Fourth, a slower decline in eGFR, with a between-group difference of 1.52 mL/min/1.73 m^2^ per year versus standard treatment, was maintained throughout the follow-up period, and this is coherent with a reduction of renal damage sustained over time. Last but not least, the nephroprotective effect was obtained in patients under optimal anti-RAS therapy and it was actually of greater extent when compared with the findings in the historical trials testing anti-RAS in similar patients with DKD [5,6].

In terms of CV protection, the CREDENCE trial extended to diabetic patients with low renal function the remarkable efficacy on the prevention of heart failure (−39% in the risk of hospitalization), which is a SGLT2-i class effect. More important, CREDENCE trial also documented the efficacy of canaglifozin in preventing atherosclerotic events with a 20% lower risk of CV death, myocardial infarction, or stroke. This finding is similar to that reported in the CANVAS, but it is at variance with results obtained in EMPAREG and DECLARE. Whether this discrepancy is related to the drug or merely due to different patient population remains to be elucidated.

Emerging evidence suggest that SLT2-i could exert beneficial effects on CV system through a significant reduction in epicardial adipose tissue, a novel, early, non-invasive biomarker and indicator of CV outcome (Table 3) [46,47,48,49,50]. This favorable effect could probably be due to the SGLT2-i-dependent loss of body fat and reduced production of pro-inflammatory chemokines.

## 4. Adverse Effects of SGLT2 Inhibitors

SGLT2-i are associated with adverse events, including genital, and to lesser extent urinary tract, infections [40], acute kidney injury (AKI), diabetic ketoacidosis, bone fracture and lower limb amputation [51].

### 4.1. Genito-Urinary Tract Infection

SGLT2-i increased by 3 times the risk of genital infections in a large population-based study, particularly in elderly subjects [52]. This finding was confirmed for all the SGLT2-i [53,54,55,56]. Therefore, patients treated with SGLT2-i should be educated to maintain appropriate personal hygiene and adequate daily water intake to prevent infections of genito-urinary tract.

### 4.2. Lower Limb Amputations

It is still unclear whether SGLT2-i increase the risk of lower limb amputation. In CANVAS, canagliflozin was associated with 2-fold higher risk of amputations (toes, feet, legs) as compared with placebo (6.3 vs. 3.4 per 1000 patient-years; HR: 1.97; 95% CI, 1.41–2.75) [38]. Conversely, EMPAREG and DECLARE did not show any risk [35,40]. However, amputation analysis in EMPAREG was a post-hoc study [57]. On the other hand, rates of amputation were low and similar to that observed with other new antidiabetic agents in a large retrospective analysis [58]. No evidence of increased risk of amputation was also detected in a large number of new SGLT2-i users (118,018 overall; 62% canaglifozin treated) versus new users of other antidiabetic oral agents [59]. The mechanisms by which SGLT2-i may increase the risk of amputation remain matter of speculation. It is possible that these drugs, by promoting volume depletion and hemoconcentration, lead to peripheral ischemia. A recent post-hoc analysis of CANVAS evidenced that risk factors for amputation were prior history of amputation, peripheral vascular disease and neuropathy; however, the authors did not identify any specific etiological mechanism [60]. Regulatory agencies have issued a warning related to this risk.

### 4.3. Bone Fractures

CANVAS has also prompted attention on the risk of fractures [38] by reporting an increase in nonvertebral fractures (HR: 1.56; 95%CI, 1.18–2.06) but not confirmed in the substudy CANVAS-R (HR: 0.76; CI, 0.52–1.12). Fralick and colleagues failed to show evidence of increased fracture risk in persons receiving canagliflozin versus a glucagon-like peptide-1 (GLP-1) agonist two nationwide US health insurance databases [61]. A similar analysis using nationwide registers in Sweden and Denmark also failed to show a significant increase in fractures in 17,213 new users of SGLT2-i versus propensity score-matched new users of GLP-1 agonists (HR: 1.11; 95% CI, 0.93–1.33) [62]. Finally, The Health Improvement Network compared 4548 persons receiving dapagliflozin with 18,070 patients under standard antihyperglycemic agents after matching for age, sex, body mass index, and diabetes duration [63]. The study did not demonstrate differences in the risk for fragility fractures (HR: 0.90; 95% CI, 0.59–1.39). Although these studies add evidence of a small (if any) risk in patients at low-risk of fracture, further evaluation is needed to determine whether these results can be extended to patients at high fracture risk (advanced age, very low bone mineral density, prior fracture, frailty, or a combination of these factors). Notably, SGLT2-i may predispose to dehydration and increased risk for falls; therefore, caution must be still used when prescribing this class of drugs to older population [64].

In a recent post-hoc analysis of a small trial in 34 patients, dapagliflozin (10 mg/d) was shown to significantly increase (by 11%) serum phosphate at 6-weeks after the start of treatment [65]. Similarly, the authors found that PTH and FGF-23 levels increased by 15% and 20%, respectively, thus hypothesizing that SGLT2 inhibition could promote phosphate reabsorption in the proximal tubule (sodium-phosphate cotransport). This effect of dapagliflozin on phosphate homeostasis may theoretically attenuate the cardiorenal benefits of SGLT2-i. The ongoing large clinical trials in the CKD setting will verify this hypothesis.

### 4.4. Diabetic Ketoacidosis

Euglycemic diabetic ketoacidosis (euDKA) can be associated with SGLT2-i in diabetes type 1 (more frequently) and type 2 (less frequently). Potential contributors are concurrent mild infections, decreased calories intake with insulin dose reduction or withdrawal, surgery. The overall risk for developing euDKA, initially described in a case series including 15 patients [66], is still unknown and ongoing trials should further elucidate this issue. In that case-series, most patients were unaware they had ketoacidosis even because, at variance with type 1 diabetes in which DKA is typically associated with severe hyperglycemia, they developed DKA in the presence of controlled glycemia [66]. SGLT2-i may induce euDKA by increasing glucose loss that in turn diminishes insulin secretion. In addition, a main determinant is the SGLT2-i related-hyperglucagonemia that per se increases the propensity toward ketone production [32,33,67]. Finally, SGLT2-i-dependent hypovolemia may also predispose to increased synthesis of glucagon, cortisol, and epinephrine, which, in turn, leads to insulin resistance, lipolysis, and ketogenesis [66]. Therefore, in patients under SGLT2-i, it is correct to test urine and/or plasma ketones at beginning of therapy and repeat testing in the case of nausea, vomiting, shortness of breath, or malaise, independently of glycemia level.

### 4.5. Acute Kidney Injury

AKI may be due to volume depletion not recognized during therapy with SGLT2-i. The FDA recommends monitoring kidney function before and after drug initiation, especially if SGLT2-i are used with other medications potentially linked to AKI, such as anti-RAS, diuretics, and nonsteroidal anti-inflammatory drugs (NSAIDs); however, SGLT2-i trials do not describe AKI as a consistent adverse event despite most patients had concomitant treatment with anti-RAS drugs and diuretics. Alternatively, SGLT2-i may be linked directly or indirectly to acute nephrotoxicity. SGLT2-i increase uricosuria because of glucose exchange with uric acid in the proximal tubule [68]. Therefore, the higher intratubular levels of uric acid may produce crystal deposition, inflammation and oxidative stress [69]. SGLT2 inhibition is also associated with shift of oxygenation from medulla to cortex; this effect may predispose DM2 patients to AKI when they are exposed to additional injuries, such as contrast media, NSAIDs, volume depletion [70,71,72]. Hence, an accurate evaluation of volemic status and screening of potential kidney toxins should be a preliminary step in patients undergoing SGLT2-i therapy.

### 4.6. Fournier Gangrene

Use of SGLT2-i has recently been associated with Fournier gangrene (FG), a rare urologic emergency characterized by necrotizing infection of the external genitalia, perineum, and perianal region. The FDA identified 55 unique cases of FG in patients receiving SGLT2 inhibitors between 2013 and 2019, with all patients treated by surgical debridement and severely ill. Even if there are no data to establish causality or incidence, physicians should be aware of this complication and to recognize it in its early stages [73].

### 4.7. Adverse Events in the CREDENCE Population

CREDENCE trial adds knowledge of the safety of canaglifozin in the CKD setting [46]. A novel finding related to occurrence of adverse events. In canaglifozin-treated patients when compared to placebo group, both amputation and fracture rates were lower as well as AKI events. On the other hand, investigators reported a significantly higher risk of diabetic ketoacidosis and genital mycotic infections in the canaglifozin arm versus placebo although either event was observed in a very small number of patients. Noteworthy, this safety profile is remarkable being observed in the DKD population considered at high risk of iatrogenic events.

## 5. Conclusions

SGLT2-i show great promise for prevention and treatment of DKD and it is likely that ongoing trials will confirm the high expectations of nephrologists as of other specialists involved in the care of diabetic patients. Available trials, and in particular the most recent CREDENCE study, have demonstrated a reduction in all-cause and CV mortality in DKD patients with DM2 and nephroprotective efficacy with a significant reduction of albuminuria and slowing of CKD progression. Intriguingly, SGLT-2-i may play a role in attenuating insulin resistance, a main CV risk factor, in diabetic as non-diabetic CKD. On the other hand, the safety profile of this drug class indicates that referral to a nephrologist is needed to monitor volemic status and to prevent AKI. Further studies are needed to verify efficacy and safety of the combination of SGLT2-i with other anti-diabetic nephroprotective drugs, namely DPP-4 inhibitors and GLP1 receptor agonists [74,75,76].

Figure 2 describes a potentially useful therapeutic algorithm for optimal control of hyperglycemia in DKD patients with mild to moderate renal impairment. This approach, which is coherent with the recommendations recently issued by the American Diabetes Association [77,78], is based on interventions that are insulin sensitizing, to limit the CV risk associated with excessive insulin dosing, and nephroprotective. More studies are needed to verify whether the choice of SGLT2-i or GLP1 agonist should be based on the presence of a specific CV disease. At the moment, balance favors SGLT2-i in heart failure and GLP1 agonist in the case of atherosclerotic disease.

## Figures and Tables

**Figure 1 medicina-55-00268-f001:**
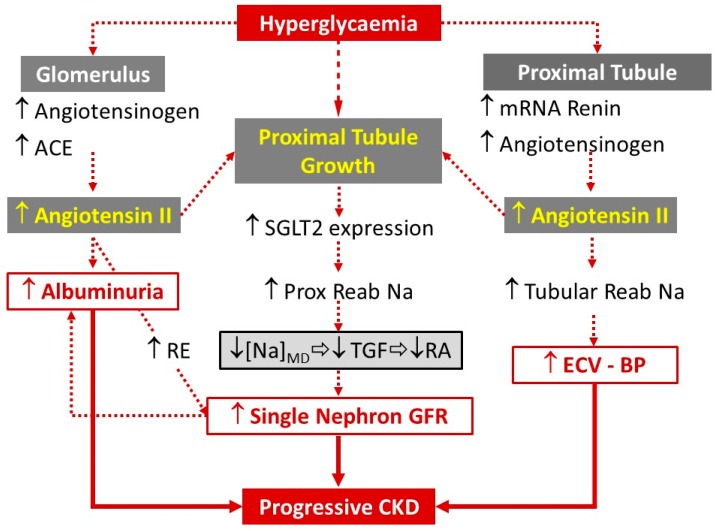
Intrarenal mechanisms underlying Diabetic Kidney Disease. Hyperglycemia increases Angiotensin II synthesis at glomerular and renal proximal tubular level with consequent proximal tubule growth (hyperplasia and hypertrophy). Hyperglycemia and tubular growth cause enhanced expression of SGLT2s with consequent higher proximal tubular reabsorption of glucose and Na. The dependent reduction of distal Na delivery to macula densa determines the deactivation of tubulo-glomerular feedback, which, in turn, allows single-nephron GFR to increase (diabetic hyperfiltration). GFR increase, abnormal albuminuria and extracellular volume expansion with dependent systemic hypertension lead to progressive diabetic kidney disease. Prox, proximal; Reab, reabsorption; RE, efferent arteriole resistance; RA, afferent arteriole resistance; MD, macula densa; TGF, tubulo-glomerular feedback; EC, extracellular volume; BP, blood pressure; GFR, glomerular filtration rate; CKD, Chronic Kidney Disease.

**Figure 2 medicina-55-00268-f002:**
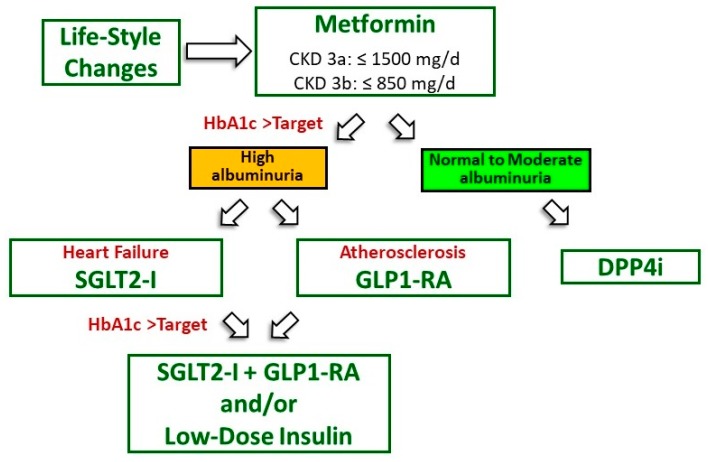
Insulin-sparing and nephroprotective antidiabetic therapy in diabetic kidney disease (eGFR > 30 mL/min/1.73 m^2^). eGFR, estimated glomerular filtration rate; CKD, chronic kidney disease; DPP4i, dipeptidyl peptidase 4 inhibitors; SGLT2-i, sodium glucose transporter 2 inhibitors; GLP1-RA, GLP1 receptor agonists. HbA1C target should be around 7%, more or less stringent according to patient features [77].

**Table 1 medicina-55-00268-t001:** Trials testing the effects of SGLT2-i on renal survival.

Trial (Sample Size)	Intervention	RAS-i Use (%)	Follow-Up (yrs)	Inclusion Criteria	Patients with eGFR < 60 (%)	Primary Outcome
EMPA-REG (*n* = 7021)	Empagliflozin/Placebo	81.0	3.1	Established CVD	26.0	3-point MACE
CANVAS (*n* = 10,142)	Canagliflozin/Placebo	80.2	2.4	Symptomatic CVD (>30 years) or two or more CV risk factors (>50 years)	25.0	3-point MACE
DECLARE (*n* = 17,160)	Dapagliflozin/Placebo	81.3	4.2	CVD and multiple risk factors for CV disease	7.0	3-point MACE
CREDENCE (*n* = 4401)	Canagliflozin/Placebo	99.9	2.6	DM2 and eGFR ≥ 30 to <90 mL/min/1.73 m^2^ and ACR > 300 to ≤5000 mg/g	59.8	composite of ESRD (dialysis, transplantation, or sustained estimated GFR of <15 mL/min/1.73 m^2^), doubling of the serum creatinine, or death from renal or cardiovascular causes

RAS-i, inhibitors of renin angiotensin system; CVD, cardiovascular disease; DM2, type 2 diabetes mellitus; eGFR, estimated glomerular filtration rate; ACR, urinary albumin to creatinine ratio; MACE: major adverse cardiovascular events, that is, cardiovascular death, myocardial infarction, or ischemic stroke.

**Table 2 medicina-55-00268-t002:** Renal outcomes in SGLT2-i trials.

Trial and Sample Size	Renal Endpoint HR (95% CI) (Drug vs. Placebo)	Antialbuminuric Effect
EMPA-REG (*n* = 7020)	0.61 (0.53–0.70) *P* < 0.001	Risk of progression to macroalbuminuria was less in empagliozin: HR: 0.62 (95% CI, 0.54–0.72) *P* < 0.001
CANVAS (*n* = 10,142)	0.53 (0.33–0.84) *P* = 0.007	Risk of progression to macroalbuminuria was less in canaglifozin: HR: 0.58 (0.50–0.68) *P* < 0.001
DECLARE (*n* = 17,160)	0.76 (0.67–0.87) *P* < 0.001	NA
CREDENCE (*n* = 4,401)	0.70 (0.59–0.82) *P* < 0.0001	Geometric mean of ACR was lower by 31% (95% CI, 26–35) during FU in the canagliflozin group

NA, not assessed; HR, hazard ratio; CI, confidence interval; ACR, urinary albumin to creatinine ratio; FU, follow-up.

**Table 3 medicina-55-00268-t003:** Effect on epicardial adipose tissue with SGLT2-i.

Study and Sample Size	Study Drug	Effect on Epicardial Adipose Tissue
Bouchi et al. (*n* = 19) [47]	Luseoglifozin	Reduction of epicardial fat volume; median decrease in EAT volume from 117 to 111 cm^3^
Fukuda et al. (*n* = 9) [48]	Ipraglifozin	Reduction of epicardial fat volume; median decrease in EAT volume from 102 to 89 cm^3^
Sato et al. (*n* = 40) [49]	Dapaglifozin	Reduction of EAT volume of 16.4 cm^3^
Yagi et al. (n = 13) [50]	Canaglifozin	Decrease in EAT thickness from 9.3 to 7.3 mm

EAT, epicardial adipose tissue.

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
