# Peer review of "SGLT2 Inhibitors: Nephroprotective Efficacy and Side Effects"

_medicina, 2019, doi:10.3390/medicina55060268_

Reviewer 1 Report

Garofalo et al., have presented a review on the ‘SGLT2 inhibitors: Nephroprotective Efficacy and Side Effects’.

The review is overall well written and included most of the latest literature.

Following are my comments,

1.Introduction:

Well presented but with numerous grammatical mistakes. Please have it reviewed by an English language expert.

2.Pathophysiology:

-Paragraph 1 on page 3 of 12 doesn’t follow paragraph 1 on page 2 of 12 and is confusing with unrelated statements. Please review.

-Please discuss how the SGLT-2 inhibitors work before paragraph 1 on page 3 of 12.

-The authors included the following in paragraph 3 of page 3 of 12.

‘Of note, CKD per se, that is, even in the absence of diabetes, leads to insulin resistance; in

 fact, CKD is a state of heightened inflammation with elevated levels of pro-inflammatory cytokines,

 and it is also associated with metabolic acidosis, excessive aldosterone, and angiotensin II levels,

 accumulation of urea and uremic toxin, that is, all factors that promote insulin resistance [35].’

I don’t think it’s necessary and can be omitted as it implies to use the SGLT2-i in non-diabetic CKD patients.

3. SGLT2-inhibitors for cardio-renal protection in T2D: Randomized Controlled Trials  

-Please include the role of SGLT2-i on the reduction of the epicardial adipose tissue (EAT) which is evaluated as a biomarker of cardiac disease in CKD patients (PMID-30832377)

Can include a table to discuss the role of SGLT2-I on EAT as discussed in the above manuscript (PMID-30832377).

4. Adverse effects of SGLT2-inhibitors

-Please include the recent warning of the Fournier gangrene risk as issued by the FDA.

Author Response

Response to Reviewer 1 Comments

- Garofalo et al., have presented a review on the ‘SGLT2 inhibitors: Nephroprotective Efficacy and Side Effects’. The review is overall well written and included most of the latest literature.

We are grateful to the Reviewer for this comment.

-1. Introduction: well presented but with numerous grammatical mistakes. Please have it reviewed by an English language expert.

We are grateful to the Reviewer for this comment. An English language expert performed a complete revision of manuscript.

-2. Pathophysiology: Paragraph 1 on page 3 of 12 doesn’t follow paragraph 1 on page 2 of 12 and is confusing with unrelated statements. Please review.

We thank the Reviewer for his comment; we have reviewed the two paragraphs accordingly.

- Please discuss how the SGLT-2 inhibitors work before paragraph 1 on page 3 of 12

We added a new sentence to explain the SGLT2-inhibitors action.

-The authors included the following in paragraph 3 of page 3 of 12: ‘Of note, CKD per se, that is, even in the absence of diabetes, leads to insulin resistance; in fact, CKD is a state of heightened inflammation with elevated levels of pro-inflammatory cytokines, and it is also associated with metabolic acidosis, excessive aldosterone, and angiotensin II levels, accumulation of urea and uremic toxin, that is, all factors that promote insulin resistance [35].’ I don’t think it’s necessary and can be omitted as it implies to use the SGLT2-i in non-diabetic CKD patients.

We removed the paragraph accordingly.

-3. SGLT2-inhibitors for cardio-renal protection in T2D: Randomized Controlled Trials Please include the role of SGLT2-i on the reduction of the epicardial adipose tissue (EAT) which is evaluated as a biomarker of cardiac disease in CKD patients (PMID-30832377)

Can include a table to discuss the role of SGLT2-I on EAT as discussed in the above manuscript (PMID-30832377).

We are grateful to the Reviewer for this comment. We added a paragraph describing the role of SGLT2-i on reduction of epicardial adipose tissue and a table on studies evaluating this effect.

4. Adverse effects of SGLT2-inhibitors. -Please include the recent warning of the Fournier gangrene risk as issued by the FDA.

We are grateful to the Reviewer for this comment. We added a paragraph on Fournier gangrene risk.

Reviewer 2 Report

Thank you for submitting your article. Parts of this review article flow very well, and others have small English usage errors that make your statements unclear.

For example, on page 1, line 41-42, you state that RAS inhibitors were introduced in clinical practice at the beginning of 2000 years.

It would be better to clearly state either the precise year (2000-2001) or that the RAS inhibitors were introduced almost 20 years ago.

Both ways of writing this sentence make it clearer that the chosen wording. 

Author Response

Response to Reviewer 2 Comments

- Thank you for submitting your article. Parts of this review article flow very well, and others have small English usage errors that make your statements unclear. For example, on page 1, line 41-42, you state that RAS inhibitors were introduced in clinical practice at the beginning of 2000 years. It would be better to clearly state either the precise year (2000-2001) or that the RAS inhibitors were introduced almost 20 years ago. Both ways of writing this sentence make it clearer that the chosen wording.

We are grateful to the Reviewer for this comment. An English language expert performed a complete revision of manuscript. In particular, the sentence quoted above has been, accordingly, changed.

Round  2

Reviewer 1 Report

I had reviewed the manuscript before and made comments and suggestions, which the authors have addressed since.